# Spatial contextual effects in primary visual cortex limit feature representation under crowding

Christopher A. Henry [1✉] & Adam Kohn[1,2,3]

Crowding is a profound loss of discriminability of visual features, when a target stimulus is surrounded by distractors. Numerous studies of human perception have characterized how crowding depends on the properties of a visual display. Yet, there is limited understanding of how and where stimulus information is lost in the visual system under crowding. Here, we show that macaque monkeys exhibit perceptual crowding for target orientation that is similar to humans. We then record from neuronal populations in monkey primary visual cortex (V1). These populations show an appreciable loss of information about target orientation in the presence of distractors, due both to divisive and additive modulation of responses to targets by distractors. Our results show that spatial contextual effects in V1 limit the discriminability of visual features and can contribute substantively to crowding.

[1] Dominick P. Purpura Department of Neuroscience, Albert Einstein College of Medicine, Bronx, NY 10461, USA. [2] Department of Ophthalmology and Visual Sciences, Albert Einstein College of Medicine, Bronx, NY 10461, USA. [3] Department of Systems and Computational Biology, Albert Einstein College of Medicine, Bronx, NY 10461, USA. ✉email: christopher.henry@einstein.yu.edu

Visual objects rarely appear in isolation. They are usually surrounded by other visual stimuli. The presence of these other stimuli—distractors—can severely impair our ability to identify a target object or distinguish its features, a phenomenon known as crowding. Crowding is a fundamental bottleneck in vision, and offers a powerful paradigm for understanding the factors that limit perceptual performance. Further, understanding crowding is important because of its central contribution to visual disorders such as dyslexia[1] and amblyopia[2].

Psychophysical studies have provided a rich description of how the parameters of a visual display influence the strength of crowding[3–6]. For instance, crowding requires distractors to be near the target, with the critical distance for crowding scaling with target eccentricity[7]. Importantly, the display properties that produce strongest crowding can be different from those that produce the most potent lateral masking, in which target detectability is impaired[8–10]. Under crowding, targets do not disappear; rather, their features become difficult to discern.

Why does crowding occur? Extensive perceptual work has led to several proposals. Some have suggested that crowding arises from interactions between the representation of targets and distractors in primary visual cortex (V1)[11,12]. Others have attributed crowding to fluctuations in attention[13]. Still others have argued that crowding is an unavoidable consequence of spatial integration, an integral component of hierarchical visual processing, which builds a representation of visual objects by combining features across space. Spatial integration has been suggested to involve pooling or compulsory averaging of targets and distractors[14,15], erroneous substitution of distractors for targets[16], the representation of the visual scene by a set of summary statistics[17,18], or some combination of these[11,16]. These forms of spatial integration have, alternatively, been attributed to the larger spatial receptive fields of individual V2[18] or V4 neurons[3].

Understanding crowding requires knowing where it arises in the visual system, how it is manifest in neuronal responses, and which neural computations and mechanisms are responsible. To date, most attempts to address the neural basis of crowding have relied on coarse measures of brain activity—fMRI or EEG recordings—which have provided equivocal answers. Crowding has been linked to modulation of V1 activity[19,20], activity distributed across the visual hierarchy[21,22] (or inter-areal interactions[23]), or activity in higher visual cortex[24,25]. Almost invariably the neural correlate of crowding is assumed to be a reduced response to a target stimulus when it is paired with distractors. However, weaker responses need not result in worse discriminability. For instance, if distractors sharpened neuronal tuning (as when a stimulus is enlarged[26,27]), summed neural responses to targets would be weaker, but encoding of their features might be more accurate.

Although sensory information is encoded and transmitted by neuronal population spiking responses, we do not know how these responses are affected by crowded displays. Previous work has shown that the responses of individual neurons to a target stimulus can be suppressed or facilitated by spatial context[28]. But it is difficult to stitch this knowledge together to gain understanding of population information. First, single neuron studies optimize stimuli for each cell separately, including centering stimuli within the receptive field. In crowded displays, most active neurons will be driven suboptimally, with varying receptive field alignment to targets and distractors. Second, population information is influenced not only by responsivity, but also by neuronal selectivity and response variability and covariability (or 'noise' correlations[29,30]). How spatial context affects these aspects of neuronal response is not well understood.

To understand how displays that induce crowding affect cortical encoding, we recorded the activity of neuronal populations in anesthetized macaque V1 to visual targets in isolation, as well as with distractors that induce perceptual crowding. We targeted V1 because of the rich neurophysiological characterization of contextual modulation there, because of its disputed role in crowding in human studies, and because V1 effects would influence and constrain the rest of the visual processing stream. To understand how changes in V1 neuronal population information compare to perceptual crowding, we complemented our recordings with psychophysical measurements in humans and monkeys. We focused on the encoding of grating orientation, a well-studied aspect of V1 function[31,32] and of perceptual crowding[11,14,24,33].

## Results

**Perceptual crowding**. Crowding depends on the parameters of a visual display, including target and distractor eccentricity and spacing[3–6]. To provide a rigorous comparison between the perceptual and neuronal effects of presenting distractors, we first measured perceptual crowding with the same display used for physiological experiments (see Methods section).

Human subjects were instructed to judge whether the orientation of a brief (0.4 s) drifting target grating, shown in the lower right visual field, was more vertical or horizontal than an internal, learned 45° reference. On two-thirds of trials, targets were surrounded by eight distractor gratings (Fig. 1a), shown in two configurations of different orientations. Distractors were identical in size to targets, had the same spatial and temporal frequencies, and were oriented ±10° from the 45° discrimination boundary.

Subjects had low discrimination thresholds when targets were presented alone (Fig. 1b, c; black), but these were clearly elevated in the presence of distractors (red). Average thresholds increased from 2.42 ± 0.07° to 4.36 ± 0.77°; the threshold elevation across subjects was 1.68-fold (Fig. 1d).

To test whether macaque monkeys exhibit perceptual crowding that is similar to that of humans, we trained three monkeys to perform the same orientation discrimination task. Monkeys had small discrimination thresholds for targets alone (Fig. 1c, middle; 2.82 ± 0.60°), and these were significantly elevated with distractors (Fig. 1d; to 5.04 ± 1.50°, a 1.59-fold increase).

Because of other experimental goals, the stimulus configuration in monkey perceptual experiments differed slightly from those used in the neuronal recordings reported below (and human experiments above). Primarily, we used four distractors in these experiments and a briefer stimulus presentation. To allow a direct comparison of human and monkey perceptual effects of crowding, we therefore ran two human subjects on stimulus configurations matched to those used for monkey psychophysics. These displays also produced a clear threshold elevation under crowding (Fig. 1c, d, right; 1.27-fold increase).

We conclude that both humans and macaque monkeys exhibit crowding for our displays (see ref. [34], for a similar comparison for displays of letters). Crowding was manifest as an elevation of discrimination threshold, which varied from roughly 25% to 70% depending on the subject and precise details of the visual display.

**Distractors affect information about target orientation**. We next assessed how distractors affect the encoding of target orientation in V1. We carried out extracellular recordings from neuronal populations of six anesthetized macaque monkeys (distinct from those trained on the perceptual task). Arrays of 1 mm electrodes were implanted to a nominal depth of ~600 microns, targeting layers 2/3 and 4B. Signals were sorted offline into single-unit and multi-unit activity. Effects were similar for the two types of recordings, so the data were pooled.

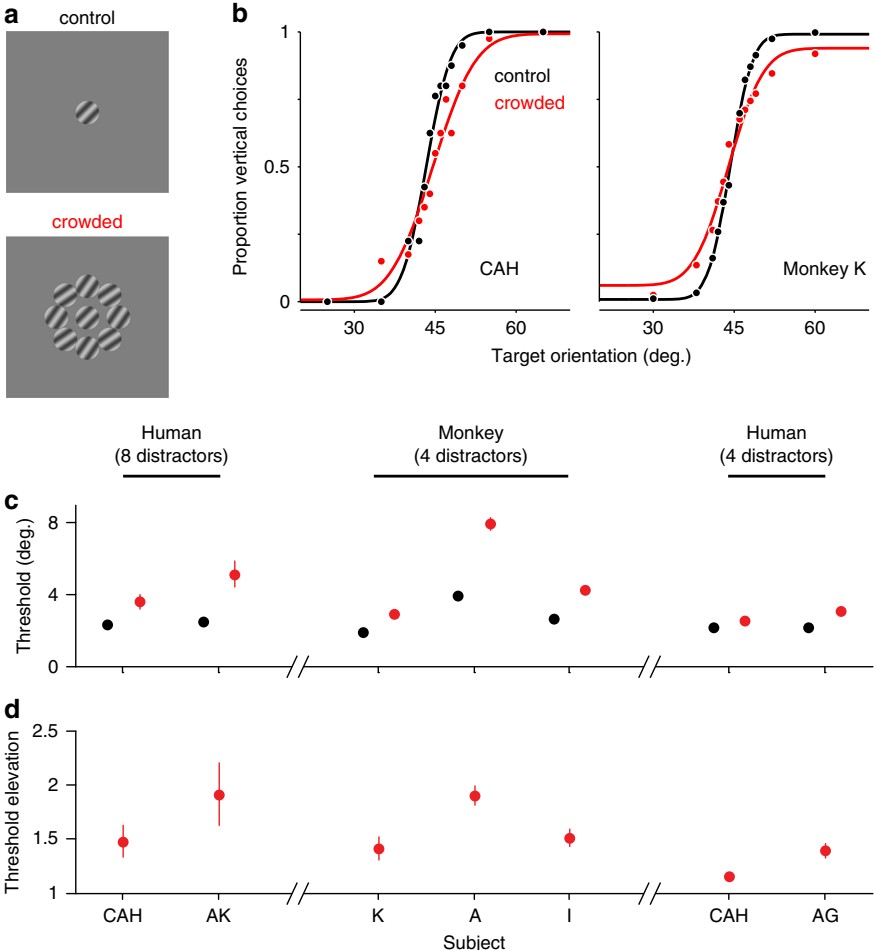

**Fig. 1 Psychophysical measures of crowding. a** Example task displays, for targets alone (top) and with distractors (bottom). **b** Example psychometric functions for one human (left) and one monkey observer (right), for targets alone (black) and with distractors (red). Curves are cumulative normal fits to the data. **c** Subjects' discrimination thresholds for targets alone (black) and with distractors (red; all points represent mean ± s.e.m. across sessions). Left: human performance for stimuli matched to physiological experiments. Center: monkey performance on similar stimuli (see Methods section). Right: human performance on stimulus parameters matched to those used for monkeys. **d** Threshold elevations under crowding (ratio of thresholds with distractors compared to targets alone; points represent geometric mean ± s.e.m. across sessions). Source data are provided as a source data file.

For each array implant, we first mapped the spatial receptive field of each sampled neuron, and then centered all subsequent stimuli over the aggregate receptive field of the population (Fig. 2a). Receptive fields were located in the lower visual field, at an eccentricity of 3–5°.

On a third of trials, we presented target gratings (0.8° diameter) alone. Target orientation was varied over a 20° range (spacing of 2–4°) across trials, spanning a small part of the orientation tuning of each cell (Fig. 2b, shaded gray region). On remaining trials, we presented targets with distractors, in two configurations (as in perceptual experiments). Distractors suppressed responses to target gratings in some cells (Fig. 2c top, black vs. red), but led to response facilitation in others (Fig. 2c, bottom right).

We quantified how distractors affected neuronal population information about target orientation using linear discriminant analysis. Specifically, we determined how well two target stimuli could be distinguished using the measured V1 neuronal population responses. Linear discriminant analysis involves finding the linear classification boundary that best separates two response distributions, shown for a sample two-neuron population in Fig. 3a (black for targets alone; red for displays with distractors). Note that the classification boundary may be different for the two types of displays (slope and intercept of the line), so we optimized this boundary separately for responses

to targets alone or with distractors. We focused on linear decoding, as is common practice[32,35], because it is biologically plausible, simply requiring appropriate choices of synaptic weights in the inputs provided to downstream 'read out' neurons[30,36,37].

We applied linear discriminant analysis to all pairwise comparisons of target orientation, using all neurons responsive to the target stimuli in each recording session (see Methods section). We compiled discrimination performance (evaluated on held out data) as a function of the orientation difference between the two target stimuli, yielding a population neurometric function. As shown for an example population (Fig. 3b), performance was better for targets presented alone (black) than for targets with distractors (red). For this population, the discrimination threshold (defined as the orientation difference resulting in 75% correct) for targets presented with distractors was 1.33-fold higher than for targets presented alone (10.9° vs. 8.2°). Across 18 populations, there was a 1.08-fold elevation of threshold under crowding (Fig. 3c, open red circles; $p = 0.01$, one-sided $t$-test).

The effect of distractors on neurometric thresholds varied substantially across populations. We found that the degree of threshold elevation was strongly related to the discrimination performance of the sampled population for the particular set of

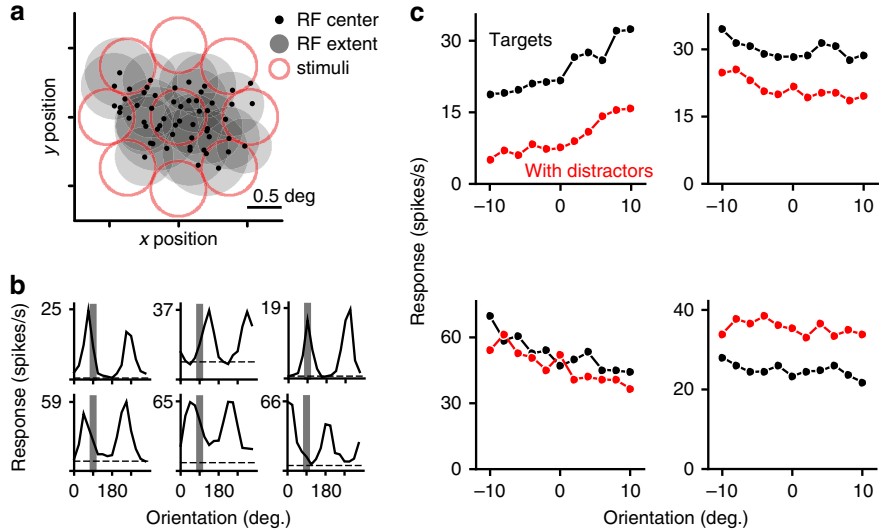

**Fig. 2 Experimental approach for V1 physiology. a** RFs of each neuron were mapped using small gratings. Black dots indicate RF centers for all neurons in one example recording; shaded gray circles indicate RF size for a subset of neurons, defined as 2 SDs of a Gaussian function fit to the responses. Red shading indicates the location of the target (center) and distractors. **b** Orientation tuning of example neurons. Shaded gray bars indicate the range of target orientations presented. Black dashed lines indicate spontaneous rate. **c** Responses of four example neurons to isolated targets of different orientations (black) and to those same targets when surrounded by distractors (red). Units are different from those shown in **b**.

target orientations presented (Fig. 3d; Pearson correlation, $r = -0.68$, $p = 0.002$). When the sampled population was informative —yielding thresholds similar to the perceptual thresholds of humans and monkeys (Fig. 1)—distractors caused a nearly 1.5-fold threshold elevation. When the population was less informative (i.e. had a high threshold), distractors caused little change in threshold. For these less informative populations, the neurometric function was not well constrained by the limited range of orientations shown, impairing our ability to estimate thresholds. We thus directly compared our ability to discriminate pairs of target stimuli separated by 8–12° of orientation, with and without distractors. In populations with thresholds >18° (mean threshold elevation: 0.97; $n = 5$), discrimination performance was significantly worse with distractors (proportion correct $0.592 \pm 0.007$ vs. $0.578 \pm 0.005$, $p = 0.02$, one-sided paired $t$-test). Together, these results suggest that distractors consistently reduce the discriminability of target orientation, and that our estimates of average threshold elevation are conservative.

The threshold elevation described above reflects an absolute loss of the information that can be extracted using a linear decoder, since the read out was optimized separately for responses to targets alone and with distractors. However, it is not clear whether signals from V1 can be read out differently for these two displays, on a moment-by-moment basis. Using instead a single decoder for displays with targets alone and with distractors is a form of suboptimal decoding, which can produce additional information loss[38].

We considered a straightforward scenario in which the shared decoder is optimized for responses to targets presented alone, and then applied to responses measured in the presence of distractors. This is one of many possible suboptimal decoders. We chose it because it represents a sensible strategy for extracting information about target orientation, except it fails to account for any response modulation evoked by distractors. When applied to our example population, the discrimination threshold for this decoder was 11.7° when distractors were present, a 1.43-fold elevation compared to the threshold for targets alone. Across populations, discrimination thresholds increased on average 1.21-fold (Fig. 3c, filled red circles; $p < 0.001$, one-sided $t$-test). Thresholds using this suboptimal decoder were significantly higher than those achieved

using a decoder optimized separately to responses to targets with distractors ($p < 0.001$, paired one-sided $t$-test).

In summary, the presentation of distractors causes an absolute loss of V1 population information about target orientation. In informative populations, thresholds can be elevated nearly 50%, accounting for a large portion of the perceptual loss of performance. Discriminability is even more strongly impaired when responses to targets with distractors are read out using the optimal decoder for targets alone. Thus, the information loss caused by distractors could be amplified by a suboptimal read out of V1 population signals by downstream circuits.

**The influence of target–distractor spacing.** Perceptual crowding is strongest when distractors are nearby, and decays as they are placed further away. Bouma's rule states that the largest separation at which crowding can be observed (referred to as the critical spacing) is roughly 0.4–0.6 times the target eccentricity[5,7,39]. We therefore asked how the population encoding of target orientation depended on target–distractor spacing, and whether there was a critical spacing for the loss of neuronal population information that was consistent with Bouma's rule.

We varied the distance between distractors and the target from 1.06° to 3.28° (center-to-center spacing; Fig. 4a), in 12 V1 populations. For each spacing, we quantified neuronal population discrimination thresholds using decoders separately optimized for responses to targets alone and with distractors. Thresholds were most elevated when distractors were nearby, and decayed gradually as spacing increased (Fig. 4b). When distractors were presented at distances >2.3°, there was no significant change in discrimination threshold ($p > 0.1$ for 3.28° offset, one-sided $t$-test). Given that the receptive fields of the sampled populations were at an eccentricity of 3–5°, this critical spacing is consistent with that predicted from Bouma's rule.

**How distractors affect neuronal responses.** We next asked what changes in neuronal responses were responsible for the neuronal population information loss with distractors. Altered neuronal responsivity, selectivity, variability, and covariability—and their interactions—can all affect population information[29,30].

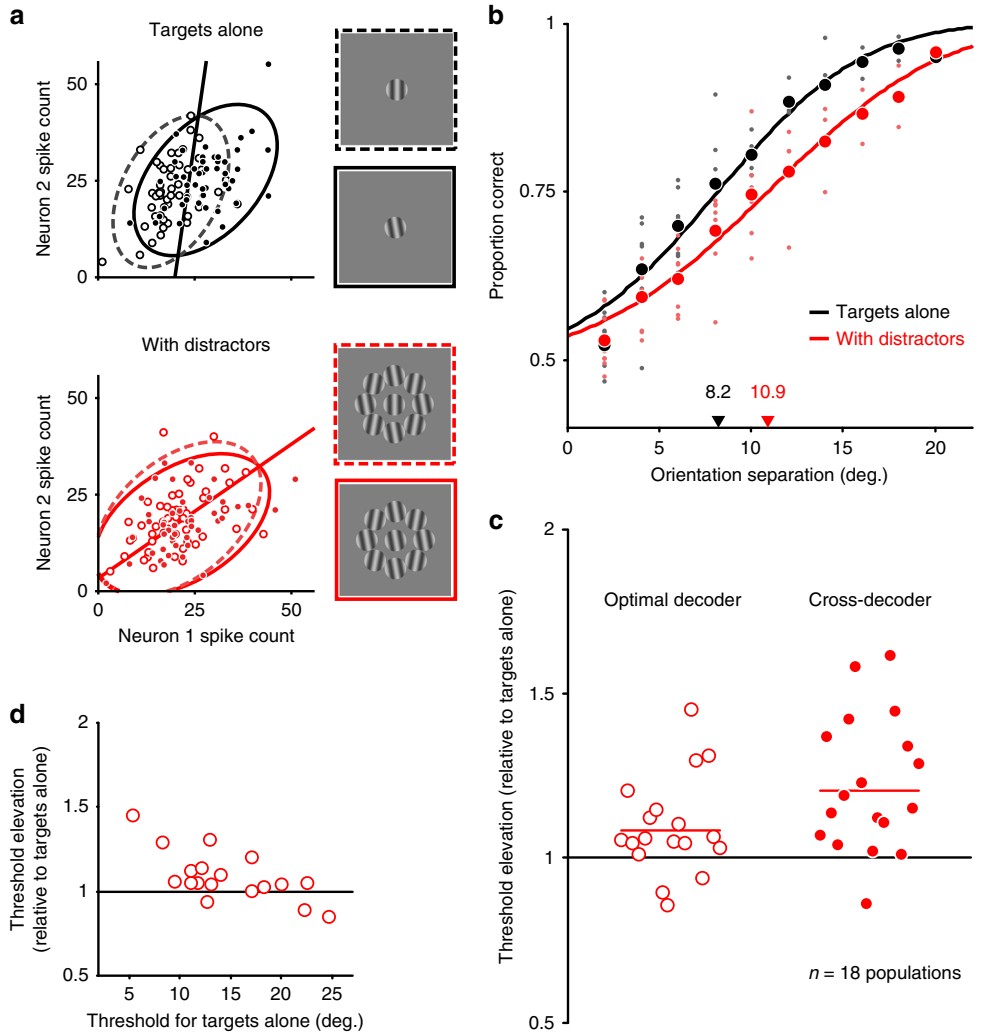

**Fig. 3 Distractors reduce target orientation information in V1 populations. a** Responses of a sample two neuron population to targets alone (black, top) and with distractors (red, bottom). Open and filled circles represent responses to two different target orientations. Ellipses indicate the 95% confidence interval of each response distribution. Linear discriminant analysis was used to identify the optimal classification boundaries to best separate the two response clouds (black and red lines). **b** Neurometric function for an example neuronal population, for targets alone (black) or with distractors (red). Each small point represents discrimination performance for one pairing of two targets of a given orientation separation. Large circles indicate the mean performance at each separation. Lines indicate fits of a cumulative normal distribution to the data. Threshold is defined as the separation producing 75% correct performance. Arrows on the abscissa indicate thresholds. **c** Orientation discrimination thresholds with distractors, relative to thresholds for targets presented alone. Each point represents one V1 population. Open circles, left: thresholds for the optimal linear classifier (points are distributed along the abscissa for visibility). Filled circles, right: thresholds when the optimal classifier for responses to isolated targets is applied to responses to targets with distractors. **d** Discrimination threshold elevation (with distractors, relative to targets alone) as a function of discrimination threshold for targets alone. Source data are provided as a source data file.

We first evaluated how distractors affected neuronal responsivity. For each neuron, we calculated a modulation index, defined as the ratio of the responses to targets with distractors relative to responses to targets presented alone. The modulation index was combined across target orientations by taking the geometric mean of the indices computed for each target. On average, distractors suppressed neuronal responses, with a mean modulation index of $0.78 \pm 0.02$ (Fig. 5a; $p < 0.001$). However, there was a broad range of modulation index values, from strong suppression in some neurons to robust facilitation in others (Fig. 5a, black bars).

The information provided by individual neurons is inversely proportional to their response variability[40,41]. To quantify how distractors affected response variability, we measured the response variance, normalized by its mean (Fano factor). There was little change in the Fano factor for responses with distractors

present compared to targets alone (Fig. 5b; geometric means: 1.946 and 1.953, respectively; geometric mean ratio, 1.00, $p = 0.72$).

To understand how changes in responsivity (including any changes in selectivity) affect single-neuron discriminability, we used receiver operating characteristic (ROC) analysis to distinguish between responses to target orientations spaced 20° apart. Values of 0 or 1 for the area under the ROC curve indicate perfect discriminability (for opposing signs of tuning), and 0.5 indicates chance performance. Single-neuron discriminability was worse with distractors, with area values closer to chance (Fig. 5c). This is evident in the slope of the linear regression, which was significantly <1 (0.78, with 95% confidence interval of [0.74 0.82]). Similarly, the deviation of area values from chance was significantly smaller with distractors ($p < 0.001$, one-sided paired $t$-test).

Moving beyond single neuron response metrics, we next considered the influence of distractors on shared variability, or pairwise noise correlations[42] ($r_{sc}$). Correlations can have a profound influence on population information[29,30,43]. We found correlations were slightly larger for responses to targets with

distractors ($0.298 \pm 0.003$) than for responses to targets alone ($0.294 \pm 0.003$, Fig. 6a; mean pairwise difference $0.005 \pm 0.001$, $p < 0.001$).

Although the modulation of correlations was on average small, even slight changes can strongly influence population information, if they are structured appropriately. To assess whether changes in correlations contribute to the information loss with distractors, we measured the neuronal population discriminability after trial shuffling the measured responses to strongly reduce correlations. For shuffled responses, distractors still led to higher thresholds compared to targets alone, as they did in the raw data (Fig. 6b, open red circles; 1.07-fold increase, $p = 0.03$, one-sided $t$-test). Thresholds for shuffled data were elevated further when we applied the optimal decoder for shuffled responses to targets alone to shuffled responses to targets with distractors (Fig. 6b, filled red circles; 1.33-fold increase, $p < 0.001$, one-sided $t$-test). If altered correlations contributed meaningfully to information loss with distractors, performance should not have worsened with distractors in the shuffled data. Because the threshold elevation with distractors was similar for raw and shuffled responses, population information loss was not caused by altered correlations.

In summary, we found that the presentation of distractors altered responsivity, reduced single neuron discriminability, and affected pairwise correlations. Analysis of shuffled responses suggests that the loss of population information with distractors is due primarily to changes in single neuron encoding.

**Modulation, spatial integration, and information loss.** Why are single neurons less informative about target orientation in the presence of distractors? We wondered whether the loss of information with distractors followed simply from the loss of responsivity. If so, information loss might occur for neurons whose receptive fields were aligned with the target grating position—and thus suppressed by distractors falling in their receptive field surround—but not for neurons with offset receptive fields. Indeed, we found that the modulation of neuronal responsivity by distractors was clearly related to the alignment of the receptive fields with the target, with aligned neurons usually suppressed (Fig. 7a, modulation index < 1) and offset neurons often facilitated (Spearman correlation: 0.28, $p < 0.001$). If information loss were driven solely by a loss of responsivity, it might be partially, or fully, offset by reading out responses from neurons with slightly offset receptive fields.

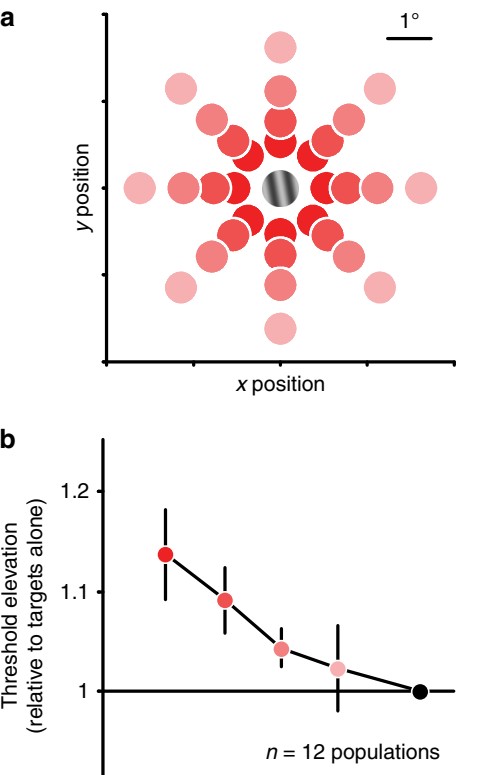

**Fig. 4 Effect of target–distractor separation. a** Across trials, we varied the spacing of targets and distractors, from distant (3.28° center-to-center distance, pink) to near (1.06°, red). Trials were randomly interleaved. **b** Average thresholds for responses to targets with distractors, normalized to thresholds for targets alone. Error bars represent s.e.m. across populations. Source data are provided as a source data file.

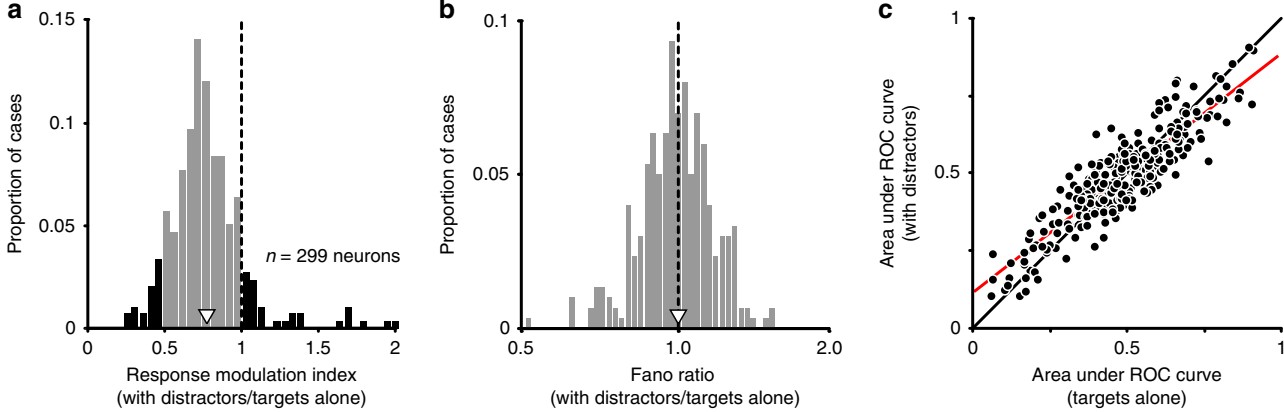

**Fig. 5 How distractors affect V1 neuronal responses to targets. a** Effect on responsivity. Histogram shows the response modulation index (the average response to targets with distractors compared to targets alone). Shaded black tails indicate cases with particularly strong suppression (index ≤ 0.5) or with facilitation (index ≥ 1). Open arrowhead indicates mean. **b** The Fano factor for targets with distractors, relative to targets alone. Open arrowhead indicates mean. **c** Single neuron discriminability (area under ROC curve) for target stimuli separated in orientation by 20 degrees, for targets alone and with distractors. Unity line shown in black; red line represents linear regression line. Source data are provided as a source data file.

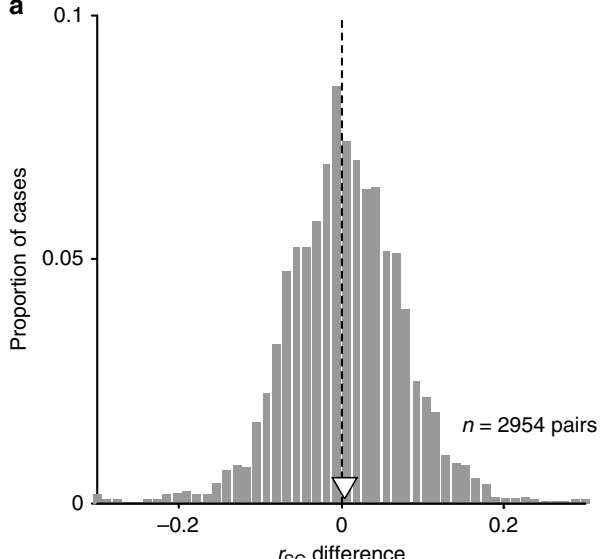

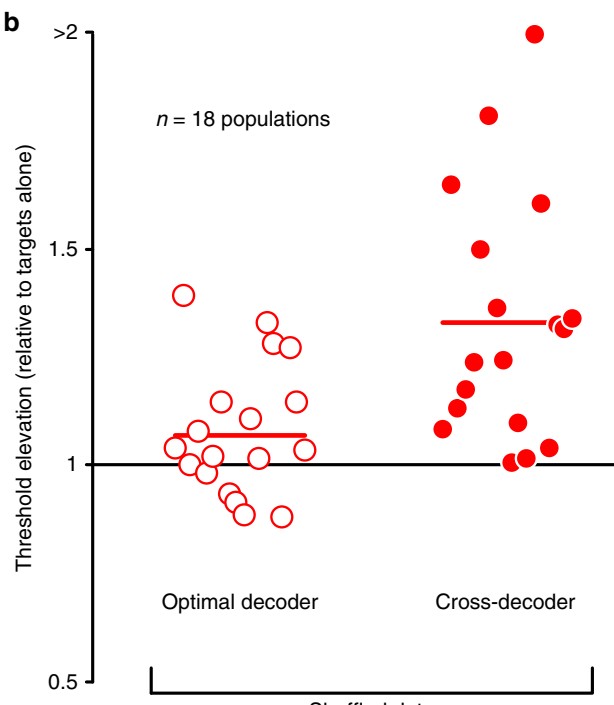

**Fig. 6 Changes in correlations do not drive population information loss. a** Changes in spike count correlation ($r_{SC}$) with the presentation of distractors ($r_{SC}$ with distractors—$r_{SC}$ for targets alone). Open arrowhead indicates mean. **b** The discrimination thresholds for each population, normalized by the thresholds for targets alone. Thresholds were measured using responses that were trial shuffled to reduce correlations. Left: thresholds for targets with distractors, determined using an optimal decoder for each display, in open red circles. Right: thresholds estimated using a decoder optimized for shuffled responses to targets alone and applied to shuffled responses to targets with distractors. Source data are provided as a source data file.

To address the relationship between response suppression and information loss, we considered separately neurons that were strongly suppressed (modulation index values ≤ 0.5, 8.4% of neurons) or moderately facilitated (values ≥ 1, 10.4% of neurons) by distractors (Fig. 5a, black bars). From these

neurons, we constructed small subpopulations. Because these subpopulations contained fewer units, performance was closer to chance, precluding an estimation of the population neurometric threshold. We therefore directly compared performance for each pairing of target stimuli, between responses to targets alone and with distractors. We excluded cases in which the population performance was not different from chance, for either condition (defined as 60% correct, which is the 95% confidence interval for chance performance given a binomial distribution and the sampled number of trials). For subpopulations of strongly suppressed cells, performance in remaining cases was worse in the presence of distractors (Fig. 7b, black points). The geometric mean performance ratio (with distractors/targets alone) was 0.92 ($p < 0.001$). For subpopulations of neurons whose responses were weakly facilitated by distractors, performance was also worse with distractors (Fig. 7c; ratio of 0.90, $p < 0.001$).

To clarify how both response suppression and facilitation reduce discriminability, we first determined whether the modulation by distractors was better described as additive or multiplicative. The two scenarios were assessed by comparing the ability of two single-parameter models to account for the measured responses (see Methods section). We found that suppressed neurons were better explained by a multiplicative model, indicating a divisive scaling of tuning by distractors (Fig. 7d; 95.9% of cases, $p < 0.001$). In contrast, facilitated neurons were better explained by an additive model (100% of cases, $p < 0.001$), indicating that their tuning underwent a rigid upward translation.

Both additive facilitation and divisive suppression would be expected to lead to information loss. This can be easily intuited by considering the Fisher information provided by each neuron[44], under the assumption of Poisson variability. Fisher information provides an upper bound on the performance of an optimal linear decoder. For individual neurons, Fisher information is proportional to the square of the tuning derivative (the signal, in the numerator), divided by the response variance (the noise, in the denominator). For divisively suppressed neurons, the divisive scaling factor is squared in the numerator, but not in denominator, resulting in lower Fisher information, or worse performance (Fig. 7e, left column). For neurons whose responses are facilitated by an additive constant, tuning slope (the numerator) is unaffected but response variance (the denominator) increases, thus also leading to lower Fisher information (Fig. 7e, right column). Note that if response modulation had instead involved subtractive suppression and multiplicative facilitation, we would not expect a loss of information about target orientation.

Because there is less information provided about target orientation both from neurons that are suppressed or facilitated by distractors (typically, those with aligned or offset RFs, respectively), we expected V1 population information loss to be largely independent of the spatial pool of neurons considered. To test this expectation, we conducted simple simulations, using populations of independent neurons with diverse tuning preferences and RF locations. Population information was quantified as the Fisher information provided by the population, for targets with distractors compared to targets alone. We explored how information would be affected by considering progressively larger neuronal pools, beginning with those units whose RFs were aligned with target stimuli and adding offset neurons by increasing the amount of spatial pooling.

Populations consisting of neurons with no surround suppression (Fig. 8, red) had a slight information loss with distractors. This effect was stronger with larger spatial pooling, owing to the integration of target and distractor signals in neurons whose

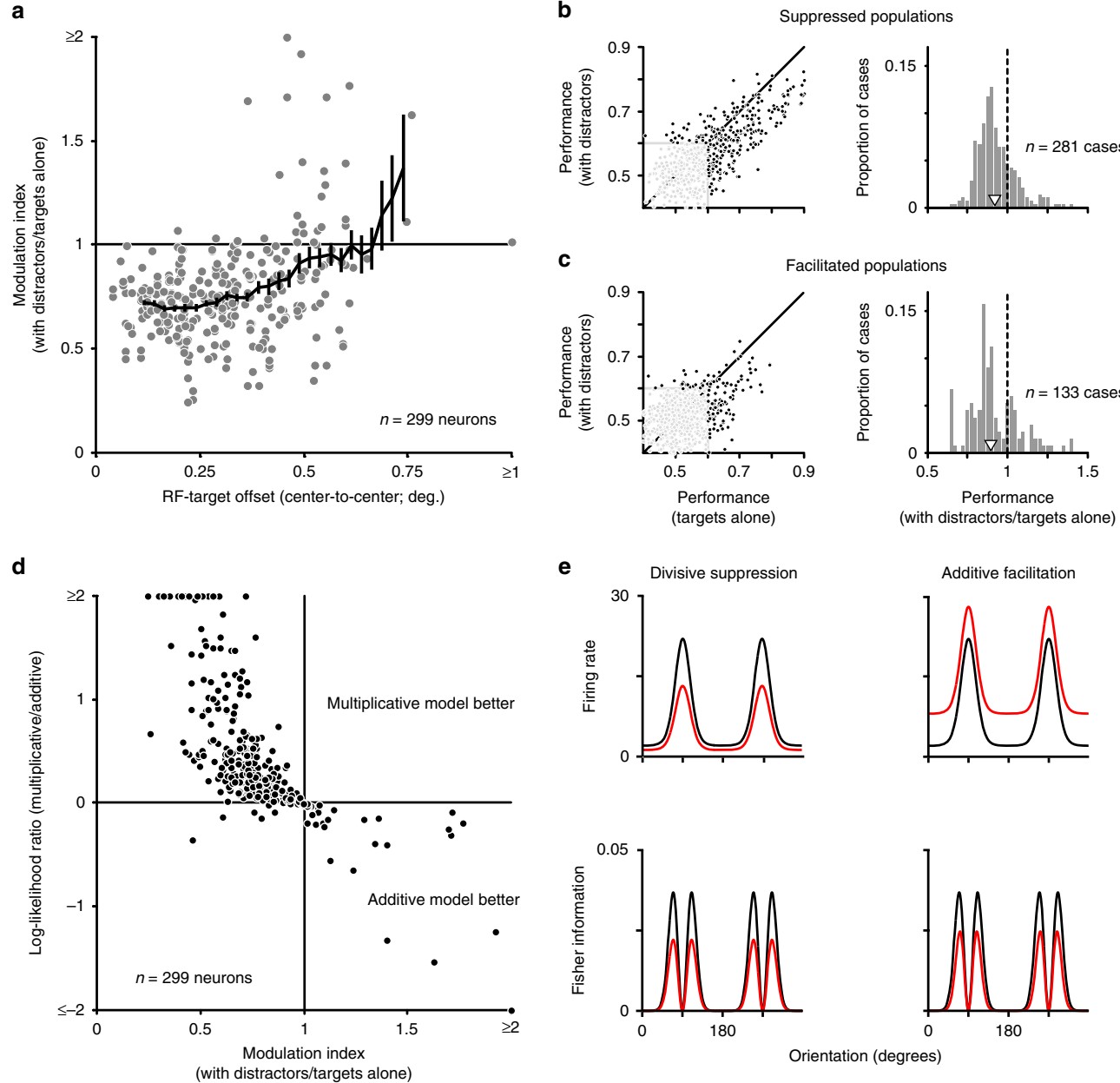

**Fig. 7 Suppressed and facilitated populations both show information loss with distractors. a** The modulation index as a function of neuronal RF offset from the target location. Black line indicates running mean, in 0.15° windows. **b** Left: discrimination performance in neuronal subpopulations with strong response suppression with distractors (modulation index ≤0.5), for targets with distractors compared to targets alone. Each point represents performance of one population to one pair of gratings. Black dots indicate cases in which performance exceeded the 95% confidence interval for chance performance for at least one condition (*n* = 281). Right: the ratio of performance for displays with distractors compared to those with targets alone. Only cases in which population performance exceeded chance are considered. Open arrowhead indicates mean. **c** Discrimination performance for neurons showing response facilitation with distractors (modulation index ≥1). 133 cases of significant performance (black dots). Conventions as in **b**. **d** Fit quality of multiplicative compared to additive models, in accounting for altered responses with distractors, as a function of the modulation index. **e** Top: tuning of model neurons, illustrating how responses for targets alone (black) are modulated by divisive suppression (red, left) and additive facilitation (red, right). Bottom: the Fisher information provided by each neuron, assuming Poisson variability. With distractors (red), Fisher information is reduced by both forms of modulation. Source data are provided as a source data file.

receptive field straddled both stimuli. When a divisive spatial surround was included (blue traces), there was greater information loss. This effect grew in proportion to the gain of the surround (blue shading), but was similar across spatial scale of pooling. Thus, the information loss in V1 responses is unlikely to be offset by considering pools of neurons with more diverse spatial receptive fields.

## Discussion

We have a rudimentary understanding of the neural mechanisms underlying crowding, a fundamental aspect of vision. Here we show that macaque monkeys experience crowding in a manner similar to humans. Further, we show that distractors reduce the information provided by V1 neuronal populations about target orientation. This information loss accounts for a significant

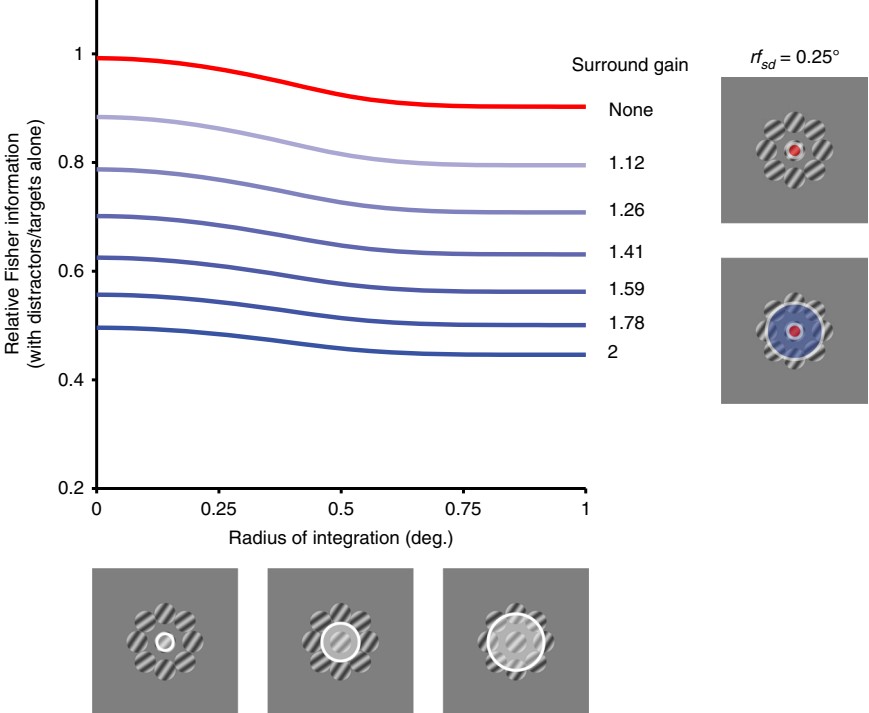

**Fig. 8 Information loss as a function of spatial scale of integration.** Fisher information (with distractors compared to targets alone) in a model population of independent neurons, as a function of increasing radius of spatial integration. Populations with excitatory receptive fields (red, s.d. = 0.25°) show a slight information loss as neurons with offset receptive fields are included. Including divisive surrounds in the receptive fields (blue, s.d. = 1.0°), leads to greater relative information loss with distractors, which scales with the gain of the surround (blue shading; values at right). Source data are provided as a source data file.

proportion of the perceptual deficit, and follows Bouma's rule, a 'hallmark' of crowding. V1 information loss is due to a divisive, suppressive modulation of neuronal responses by distractors, as well as additive summation of responses to targets and distractors in neurons whose receptive fields encompass both stimuli. Our results show that spatial contextual effects at the first stage of cortical processing place fundamental limits on perceptual discriminability of visual features.

The factors that limit neuronal population information have been studied extensively[29,30], but no previous work, to our knowledge, has assessed how spatial context influences population information. Although spatial context can affect single neuron variability[27] and its correlation across neurons[45]—both of which might alter population information—the information loss with our distractors is due primarily to a divisive reduction in single neuron responsivity and consequent loss of discriminability. An important caveat is that our populations consisted of tens of neurons. Information loss might be different in larger populations, for which information is strongly influenced by 'differential correlations'[43]. If these correlations are affected by distractors, information might be altered differently from the effects we report[30].

Neuronal response suppression by spatial context is a well-documented component of V1 processing[28], and several studies have suggested this suppression is divisive[46–48]. Neuronal surround suppression has often been linked to lateral masking, a distinct form of perceptual impairment in which target detectability is reduced by spatial context[10,49]. V1 suppression can indeed eliminate responses to low contrast targets within the receptive field[46], but it also modulates responses to high contrast stimuli. Our work shows this modulation reduces feature discriminability in suprathreshold (detectable) stimuli.

Our experiments explored a limited arrangement of targets and distractors. The distractor configurations we used produced similar loss of information and perceptual crowding (not shown), so we averaged over the effects induced by different configurations. However, crowding strength is known to depend on the properties of targets and distractors and on their spatial arrangement, and thus the effects we report might be different for more distinct distractor arrangements. Many of these dependencies of crowding on distractor configuration might seem to elude a simple mechanism like V1 surround suppression, but in fact the properties of this suppression often parallel those of crowding. For instance, V1 suppression weakens when stimuli in the receptive field and surround differ in orientation[28,50,51]. Similarly, crowding is alleviated when distractors and targets have different orientations (see ref. [3] for review). V1 suppression weakens when a stimulus in the surround exceeds a critical size[52,53]. Similarly, crowding is weaker for large distractors than small ones[11]. Finally, V1 suppression depends on the statistical dependencies between image components in the receptive field and surround, and can be 'gated' when these differ[47]. Similarly, crowding depends on larger spatial context, and is alleviated when distractors can be perceptually segmented from targets[54,55].

Though we emphasize information loss in V1, our results do not exclude the possibility that mechanisms in extrastriate cortex also contribute to crowding, particularly for complex visual features. For instance, crowding could involve interactions between targets and distractors within the larger spatial receptive field of extrastriate neurons. Such interactions have been documented in V2[56], V4[57], and IT[58]. However, this previous work did not relate these interactions to a perceptual loss of stimulus information, nor have they considered population information (except ref. [58], which considered 'pseudo-populations'). Interactions between

targets and distractors in extrastriate cortex could also involve surround suppression, which is robust in higher visual areas (e.g., V2[59,60]; V4[61]). Importantly, any degradation of extrastriate encoding can only compound the loss we observe in V1. Higher cortex is almost entirely dependent on V1 input[62]. Information loss in V1 thus limits the information provided to extrastriate cortex, which cannot encode information it does not receive.

Our results also do not exclude the possibility that crowding involves fluctuations in attentional signals[13] or that the shape of the crowding zone is influenced by saccades[63]. However, our results show that attentional mechanisms are not required to produce crowding, as our recordings were performed in anesthetized animals. Similarly, the crowding effects we measured are not influenced by eye movements, since we administered paralytics to suppress these. Importantly, V1 spatial contextual effects are robust in awake animals, and have similar properties to the modulation observed under anesthesia[64,65].

The effects of distractors on target discriminability will be strongly influenced by the strategy used to read out those signals. We found that decoders optimized separately to responses to targets alone or with distractors resulted in up to 50% information loss (in the most informative populations). Applying the optimal decoder for targets alone to responses to targets with distractors roughly doubled the effect size. There are many additional scenarios which could produce further information loss. For instance, one could decode V1 neurons whose receptive fields overlapped the distractors but not the targets. These neurons, by definition, carry no information about target orientation; decoding their responses when distractors are present could further degrade information. One could also decode different pools of neurons on different trials, leading to the erroneous decoding of task-irrelevant neurons on some trials (e.g. as in substitution[16]).

Crowding has been referred to as 'an enigma wrapped in a paradox and shrouded in a conundrum'[3]. We posit that this is due in part to a focus on establishing litmus tests for crowding. It may be more helpful to accept that there are multiple mechanisms in V1 that underlie spatial contextual modulation of stimulus detectability and identifiability. Our results show that mechanisms evident in V1, previously invoked to explain modulation of detectability, can explain much of the perceptual crowding that occurs for simple visual features like orientation. Further, our approach of assessing neuronal population information for different visual displays provides a framework for assaying the contribution of other areas and neural computations to perceptual crowding.

## Methods

**Perceptual experiments.** Stimuli were generated either with custom software based on OpenGL libraries (EXPO) or using Matlab and the Psychtoolbox extensions[66] and displayed on calibrated CRT monitors with linearized output luminance (human experiments —1280 × 1024 pixels; 85 Hz refresh rate; 57 cm viewing distance; 40 cd m$^{-2}$ mean luminance; monkey experiments—1024 × 768 pixels; 100 Hz refresh rate; 64 cm viewing distance; 40 cd m$^{-2}$ mean luminance).

Human subjects reported the orientation of a target grating presented briefly in the lower right visual field (4.24° eccentricity), either alone or with distractors. All participants provided written informed consent prior to testing, in accordance with the Declaration of Helsinki. The protocol was approved by the Institutional Review Board of the Albert Einstein College of Medicine. Stimulus parameters were identical to those used in the physiological experiments (0.8° diameter; spatial frequency, 1.5 cyc deg$^{-1}$; temporal frequency, 4 Hz; duration, 0.4 s). Distractors consisted of eight gratings spatially alternating in orientation (45 ± 10°); distractor arrays were counterbalanced across trials. The center-to-center spacing of target and distractor stimuli was 1.04°. Starting spatial phase was randomized across trials. Phase was not randomized in the physiology experiments but since we averaged neuronal responses over multiple cycles of drift, we effectively removed phase information from our decoder. Subjects were given feedback on each trial in the form of a brief sound presented following incorrect responses. We did not monitor fixation behavior in the experiments presented here. To be sure our results were not affected by poor fixation, we ran control sessions in two subjects with eye

tracking (1° window). These sessions yielded results that were indistinguishable from those reported here.

Behavioral experiments were also carried out in three cynomolgus monkeys (*Macaca fascicularis*). Eye position was monitored using a video-based eye-tracking system (SR Research), with a 1 kHz sampling rate. Animals had to maintain fixation on a central fixation point within a 1° window; after a 1 s delay, stimuli were shown for 0.25 s, at an eccentricity of 4.24°. The eccentricity was chosen to facilitate neurophysiological recordings from these animals in later sessions (not reported here); stimulus duration was chosen to facilitate neuron-choice analyses (not reported here). After stimulus offset and a further delay of 0.2 s, two choice targets appeared (above and below fixation point); animals had to saccade to the upwards target to indicate a vertical choice, downwards for horizontal, and were rewarded for correct choices with a small liquid reward. Incorrect trials were followed by a sound and brief time out.

Stimuli in monkey perceptual experiments differed slightly from those in the physiology experiments reported here, as sessions were tailored towards separate aims: gratings were 1.1° diameter, 1.0 cyc deg$^{-1}$, and 4 Hz drift rate. Distractors consisted of four gratings spaced around the target (to allow for closer spacing of distractors, without physical overlap, if this was needed to strengthen crowding). Center-to-center spacing of target and distractors was 1.43°. Distractor orientations were fixed within each behavioral session, and were offset from the 45° reference by 2–30° (median: 5°). Each distractor array consisted of two gratings of each sign of tilt; thus six distractor arrays were sampled within each session. Spatial phase was fixed to allow for neuron-choice analyses in subsequent recordings. For comparison, two human subjects performed perceptual experiments with parameters matched in every way to those used for the monkey behavior, except starting spatial phases were randomized for humans.

Perceptual performance was quantified by fitting a cumulative normal distribution (with one additional parameter for lapse rates) to the choices in each condition, via maximum likelihood. Psychometric functions were fit separately for each subject, session and experimental condition (i.e. each distractor array condition). Fitted parameter values were then averaged across conditions and sessions for each subject. Thresholds were defined using 75% correct performance as criterion. Human subjects performed 4–6 sessions, or 1920–3456 trials per subject. Monkeys performed 21–43 sessions, or 17,101–47,680 trials per animal. Lapse rates—the estimated error rate for the easiest discriminations—were low for targets presented alone (0–2% across subjects) and only slightly elevated with distractors (range 0–5%). Our quantification of thresholds accounted for these errors so that our estimates of threshold elevations were not influenced by variations in lapse rate.

**Animal surgical procedures.** Recordings were performed in six anesthetized male monkeys (*Macaca fascicularis*). Prior to surgery, animals were administered 0.01 mg kg$^{-1}$ glycopyrrolate and 1.5 mg kg$^{-1}$ diazepam. Anesthesia was induced with 10 mg kg$^{-1}$ ketamine; after intubation, anesthesia was maintained with 1.0–2.5% isofluorane (98% O$_2$/2% CO$_2$ mixture). Intravenous catheters were inserted into the saphenous vein of each leg. Animals were positioned into a stereotactic device (Kopf) and a craniotomy and a durotomy were performed over V1. A 10 × 10 microelectrode array (400 μm spacing, 1 mm length; Blackrock Microsystems) was implanted, and agar was placed over the brain to prevent dessication. During recordings, anesthesia was maintained through a venous infusion of sufentanil citrate (6–24 μg kg$^{-1}$ h$^{-1}$) in Normosol solution with dextrose. The animal was paralyzed with vecuronium bromide (0.15 mg kg$^{-1}$ h$^{-1}$) to minimize eye movements. Anesthetic depth and well being were ensured by continuous monitoring of vital signs: electrocardiogram, EEG, blood oxygen saturation, end-tidal CO$_2$ partial pressure, airway pressure, blood pressure, and temperature. Ophthalmic atropine was used for pupil dilation. Gas-permeable contact lenses protected the corneas, and external lenses were used to bring the retinal image into focus. Broad spectrum antibiotics (either 2.5 mg kg$^{-1}$ Baytril or 2.2 mg kg$^{-1}$ Ceftiofur) and an anti-inflammatory steroid (1 mg kg$^{-1}$ dexamethasone) were administered daily.

All procedures were approved by the Institutional Animal Care and Use Committee of the Albert Einstein College of Medicine and were in compliance with the guidelines set forth in the National Institutes of Health Guide for the Care and Use of Laboratory Animals.

**Visual neurophysiology.** Stimuli were generated and displayed as in the perceptual experiments, but with a viewing distance of 110 cm. Spatial receptive fields of each neuron were mapped using small drifting gratings (0.5° diameter; four orientations; 0.25 s duration) across a spatial grid spanning 3° × 3°.

Small drifting target stimuli (0.8° diameter; spatial frequency, 1.5–2 cyc deg$^{-1}$; temporal frequency, 4–6.25 Hz) were centered over the aggregate receptive field, and viewed monocularly. A reference orientation was chosen, with target stimuli spanning −10° to +10° around this value, in 2–4° steps. On two-thirds of the trials, targets were surrounded by an array of drifting distractor gratings. Distractors were presented at eight locations on a ring concentric to the target. The center-to-center spacing between target and distractors was 1.04°. Distractor orientations were set to ±10° from the reference orientation and spatially alternated in orientation to introduce heterogeneity across the distractor array. Two counterbalanced distractor arrays were used, each comprising one-third of the total trials. Target and distractor stimuli were presented for 0.4 s, with no difference in onset or offset.

Conditions were randomly interleaved across trials, with 1 s blank stimuli between trials.

We recorded in total from eight arrays implanted in eight hemispheres. In each implant, we presented two sets of target stimuli (i.e. two sets of gratings spanning a 20° range of orientation, with each set centered at a different orientation spaced 45° or 90° apart). In a few cases, we collected additional data after rotating the orientation of the entire ensemble. We considered the populations responding to each set of orientations as independent, since V1 orientation tuning meant that largely distinct sets of units would be visually driven in each case. In all analyses, we considered responses to the two sets of distractors separately, and then averaged the results.

Neural signals exceeding a user-defined threshold were digitized at 30 kHz and sorted offline into single- and multi-unit activity (Plexon Offline Sorter). Neuronal responses were measured as the total spike counts during the entire stimulus presentation (from 0 to 0.4 s after stimulus onset). Neurons with a maximal response to isolated targets that was >0.5 spikes s$^{-1}$ and 2 s.d. above the mean spontaneous response were selected for further analysis. The size of the responsive population ranged from 7 to 35 neurons (mean size: 17 neurons).

**Data analysis**. Neuronal population information about target orientation was quantified using linear discriminant analysis. For each population, the activity evoked by a target stimulus was summarized by a $T \times N$ dimensional response matrix $R$, where $T$ represented the population activity on each trial and $N$ was the number of neurons. Linear discriminant analysis finds the $N \times 1$ population weighting vector $w$, such that the projection of the population responses onto that vector ($R*w + c$) is maximally separable for responses to two target stimuli. The constant $c$ represents the criterion that optimally separates the two target response distributions within the projected subspace. Linear classifiers were trained on 48 repetitions of each target orientation and performance was measured on a separate held out test set of two trials for each target. The reported performance is the average over different folds of the data. We obtained nearly identical results when we decoded population responses using logistic regression, with Lasso regularization.

To quantify population discrimination performance, we fit an observer model to classifier performance across all pairwise combinations of target orientations. Specifically, each population performance was summarized by a cumulative normal distribution, adjusted for a two alternative forced-choice task, with parameters fit to the data via maximum likelihood. Discrimination thresholds for each population were defined as the orientation offset that produced 75% correct.

The nature of response modulation with distractors was assessed by determining whether the average response to targets with distractors, $R_{\text{with distractors}}(\theta)$, was better characterized as an additive or multiplicative scaling of responses to targets alone, $R_{\text{targets alone}}(\theta)$. Both models consisted of a single scaling parameter $c$, with the additive model as $R_{\text{with distractors}}(\theta) = R_{\text{targets alone}}(\theta) + c$, and the multiplicative model as $R_{\text{with distractors}}(\theta) = R_{\text{targets alone}}(\theta) * c$. Models were rectified at 0 to prevent negative firing rates. The log-likelihood of each model given the data was calculated as

$$\log L = \sum_\theta \log \left[ \frac{R_{\text{p}}^{R_{\text{m}}} e^{-R_{\text{p}}}}{R_{\text{m}}!} \right]$$

which is the log-likelihood of each model under the assumption of Poisson variability, given the measured and predicted responses, $R_{\text{m}}$ and $R_{\text{p}}$, respectively[67].

Spike count correlations were measured as the Pearson correlation of the spike counts of two neurons to repeated presentations of a given stimulus. We first removed trials in which the response of either neuron was >3 s.d. from its mean response, to avoid contamination by outlier responses. For both Fano factors and correlations, we calculated values separately for each target orientation and distractor configuration. The reported distributions show, for each neuron or pair, the average over the different stimuli.

**Spatial integration simulations**. Model populations consisted of independent neurons with homogenous tuning curves defined by a Von Mises distribution (spontaneous rate 3 sp/s; tuning curve amplitude 20 sp/s; bandwidth, defined as half-width at half-height, 15°). At each spatial offset from the target location (0–1°, in steps of 0.025°), the population consisted of 36 neurons with uniformly distributed orientation preferences (10° spacing). Neuronal spatial receptive fields consisted of a circular 2D Gaussian distribution (s.d. = 0.25°). Target gratings were 0.8° diameter. Neuronal responses to a given target were calculated by scaling each neuron's predicted mean response (i.e. tuning) for that orientation by the spatial overlap of the target with the receptive field. Distractors were offset by 1.04° from the target, with orientations tilted ±10° from the reference orientation. Responses to distractors were computed similarly to those for targets. The response to targets and distractors in the receptive field was the sum of the responses to each stimulus.

To incorporate surround modulation, we ran simulations in which the receptive field included a divisive surround, defined as a circular 2D Gaussian distribution with a larger spatial profile (s.d. = 1.0°). Surround activation was calculated by the spatial overlap with the stimulus array, multiplied by a gain term to set the maximal strength of the surround. Responses from the excitatory receptive field were divided by this surround activation. The surround was thus spatially selective,

but non-selective for any specific stimulus features (e.g. orientation). Population Fisher information for target orientation was calculated by summing the Fisher information from individual neuron tuning curves, defined as the square of the tuning derivative divided by the response variance.

All indications of error are standard error of the mean, unless noted otherwise. Averages over ratios are calculated as the geometric mean; ratios were log-transformed before statistical testing. All statistical tests are paired two-sided $t$-tests, except where noted.

**Reporting summary**. Further information on research design is available in the Nature Research Reporting Summary linked to this article.

## Data availability
Data are available upon request. Source data for Figs. 1 and 3–8 are provided as a source data file.

## Code availability
Code is available upon request.

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

## Acknowledgements

We thank the Kohn lab for help with experiments, and Ruben Coen-Cagli for helpful comments. This work was supported by NIH grants (EY023926 and EY028626) and the Charles H. Revson Senior Fellowship in Biomedical Science.

## Author contributions

C.A.H. and A.K. designed the project; C.A.H. performed experiments and analyzed the data; C.A.H. and A.K. wrote the paper.

## Competing interests

The authors declare no competing interests.
