## [Peer Review File · Nature Communications]

Reviewers' Comments:

Reviewer #1:

Remarks to the Author:

Review

In this manuscript the authors present a clear and concise study of the neural basis of crowding. A simple experiment was conducted in humans to demonstrate the effect, which was then repeated in monkey. Multiunit anesthetized recordings demonstrate a comparable effect within V1 that can explain many of the perceptual effects of crowding. Hypotheses were tested using well-reasoned experiments and simple statistical tests. The existing literature was heavily referenced, connecting their thorough study to multiple fields.

The authors are to be commended for producing an exceedingly well-prepared manuscript – Figures and captions are clear, the document is brief and progresses naturally. There is little if anything I would take away from the text, and it has an appropriate level of complexity for the questions asked and answered. The document appears virtually free of typographical errors, and care was taken to produce pleasing figures that effectively illustrate the multiple notable findings.

After reviewing the manuscript I have a few minor issues that could be addressed to (in my personal view) enhance the study and are not intended to be critical of the manuscript as a whole. I have a few significant concerns that I believe should be addressed, but these could simply be due to a misreading of the manuscript and resolved by clarifying the appropriate result and methods sections.

Once these issues are resolved I would be happy to recommend this article for publication.

Minor

Abstract, sentence 5: punctuation, "...visual cortex (V1), These populations show...".

Contour segmentation - Is there any relationship between distractor arrangement and target judgement? In the methods it is described that distractors are randomly set to 35deg or 55deg orientation, but one could imagine that given the fixed arrangement of the distractors, there are some combinations of distractors and targets which might induce higher-level perceptual effects – some arrangements might form curvilinear paths that are perceptually grouped into long contours, others might not. I'm just wondering if some of the difference between control and crowding might be explained by specific combinations of distractors. The authors note in the discussion that the spatial arrangement of distractors is significant to crowding, and I believe some arrangements of the distractors might in fact enhance discriminability. It seems that a simple analysis of psychophysical data collected in their experiments might capture this effect if it should exist.

Page 10, paragraph 3 sentence 2: 'They' referring to 'our results' was a little awkward to parse and could be improved.

Additive vs multiplicative modulation: The additive facilitation and divisive scaling effect is clear and quite interesting. I believe that this result may be explained by a model where the neuron's activation function is nonlinear. Specifically, suppose response $R = f(r(\theta) + c)$, where $f(x)$ approximates an exponential from $-\infty < x < f^*$, and is linear for $f^* < x < \infty$. $r(\theta)$ is the neuron's orientation preference. $R_{\text{alone}}(\theta) = f(r_{\text{alone}}(\theta) + c)$. I believe a unit of this form would produce the additive facilitation effect for $c > 0$ and divisive scaling for $c < 0$ for some tuning curve shape r . However, this is likely just semantics describing the method by which a neuron's orientation preference is mapped into an output spike rate. It might be useful to mention why it matters that facilitation is additive and suppression multiplicative, and what mechanisms could be at play. It seems like the authors want to hint at suppression being the result of surround-dependent divisive inhibition, but this

is a tenuous claim and should be avoided.

Major

Human and monkey psychophysics: eye position and feedback

I have a concern regarding the methods of measuring crowding between species. In your psychophysical methods you describe that humans had target presentations of 0.4 seconds, while monkeys had 0.25 seconds. In the results paragraph 3 and Fig1c it is described that another experiment was run for humans matched to monkey psychophysics – does this include presentation times? It might be helpful to explicitly title the left and right regions of Fig1c to separate these two conditions.

In any event, 0.25 seconds is quite long for a human judgement, long enough for a visually-guided saccade and spatial attention to shift. In the monkey case eye position was monitored, but it's unclear that human eye positions were similarly restricted. Given that eye-tracking details were given in the primate psychophysics paragraph of the methods, one can assume that eye position was not monitored for human subjects. I believe that it is necessary to ensure subjects were fixating within a 1 deg. window to eliminate the possibility of voluntary or involuntary saccades contaminating performance.

Also, it is not mentioned if humans received feedback for correct/incorrect choices – the monkeys were provided feedback by water reward, and therefore could adapt their decision strategy to maximize performance. I believe that human subjects would also need to be given feedback after every trial to ensure a fair comparison.

If either or both of these conditions were matched across humans and animals, please indicate so within the text. A difference in these conditions would need to be justified.

Phase

I am concerned about the lack of phase randomization within the psychophysical experiments. Given that the targets were presented at a fixed spatial frequency and phase, I believe it is possible for an observer to perform the task by instantaneously measuring a single pixel in the display – it being above or below 50% gray would indicate horizontal or vertical orientation. I understand that this would not be possible for an animal when viewing a target at 4-5degrees eccentricity, but I still wonder why phase wasn't randomized during monkey psychophysics as simple cues to the judgement exist in the stimuli. I feel it is necessary for the authors to justify this difference in experimental conditions.

Response dynamics

I would expect that there is an appreciable latency in the response modulation due to crowding, should the multiplicative scaling of responses be due to connections within V1. Therefore, how does the modulation of responses demonstrated in Fig7 evolve over time? If the neural response to crowding is delayed, I would expect that the V1 populations would more accurately decode orientation during the initial transient response period, and get worse as crowding effects modulate responses. It might be the case that V1 could accurately decode orientation during this initial transient period and thus not limit feature representation under crowding. I feel that it is critical for the authors to address this to rule out crowding as merely a latent effect within V1, as that would place the neural basis for crowding at a later stage of processing.

Reviewer #2:

Remarks to the Author:

Overall this is a well-written manuscript investigating how contextual effects in primary visual cortex might underlie the psychophysical phenomenon of crowding. As the authors clearly articulate, crowding has been studied extensively in human subjects but the neural basis has not been systematically investigated. This manuscript shows that monkeys, like humans, experience crowding, and details how contextual effects in V1 could at least partially underlie crowding. This manuscript and data represent a valuable, novel addition to the literature. Experiments are well-designed and the analyses are appropriate. All of my comments below are suggestions aimed at trying to improve clarity and detail.

1. It will be beneficial to the reader if the authors could provide more details in methods specifically with regard to the rationale behind parameter choice. I do appreciate these are primate experiments and the parameter space cannot be explored extensively but some more detail would be appreciated. For example, the stimulus eccentricity of 4.24 for the behavioral task – was this based on the mean RF in the monkey experiments? Is there a reason for choosing 400 ms and 8 distractors for the human studies but 250 ms and 4 distractors for the monkey? Why is target-distractor spacing 1.04 in one experiment but 1.1 in the other? It seems some of this is motivated by another study but a little more detail as to why those specific parameters were chosen would be beneficial to the reader. As of now, these seem rather arbitrary. It seems that a short paragraph explaining why these numbers were chosen would be helpful.
2. Were eye signals monitored in the experiment with the human subjects? Please provide details including window size.
3. Is it true that the distractors orientations all deviated by the same amount from 45 in the monkey task as in the human task? If so, please state explicitly.
4. In Figure 1a it would be helpful to show panels for both types of tasks (human and monkey) and then in 1c, show small versions of those corresponding to the two human and one monkey panel. As presented, the figure and legend don't provide any hints about the differences between the two human panels. Help the reader!
5. In Fig 2a it will be good to visualize the size of V1 RFs. So, the stimuli (shown in red) could simply be drawn as outlines so the V1 RFs show through.
6. How are V1 RF sizes estimated? Is it 1SD? 2SDs? Please specify.
7. Are units in Fig 2c included in 2b? If so, please identify.
8. Levi and Carney 2009 is cited as Carney and Levi on page 10. Either that, or a reference is missing. Please fix.

Reviewer #3:

Remarks to the Author:

In this study, Henry and Kohn investigate the neurophysiological underpinnings of visual crowding. Crowding is ubiquitous in peripheral vision and although much is known about the phenomenology, almost nothing is known about the underlying neural mechanisms. This study is important and is the first to directly assess the possible neural mechanisms and associated computations in the primary visual cortex (V1). The authors conclude that spatial contextual effects limit the discriminability of visual features in V1 population activity and this loss of information might provide a fundamental limiting factor for the rest of the visual processing hierarchy.

I would like to congratulate the authors for an elegantly conducted study and for a very well written manuscript.

I have no major comments.

I have two minor comments:

- the criteria that were used to estimate the discrimination thresholds from the psychometric curves is not stated
- the Nandy & Tjan (2012) study has been mischaracterized (p3 and p10) in stating that crowding can be attributed to saccadic sampling. That study explored saccadic sampling as a means to explain the radially elongated **shape** of crowding zones but was very much in alignment with the present study that crowding itself was caused by contextual effects.

Anirvan Nandy
Dept. of Neuroscience
Yale University

We thank the reviewers for their supportive remarks and helpful suggestions. We have collected new psychophysical data and performed additional analyses to address their concerns. We have also modified our manuscript as suggested.

Reviewer #1:

In this manuscript the authors present a clear and concise study of the neural basis of crowding. A simple experiment was conducted in humans to demonstrate the effect, which was then repeated in monkey. Multiunit anesthetized recordings demonstrate a comparable effect within V1 that can explain many of the perceptual effects of crowding. Hypotheses were tested using well-reasoned experiments and simple statistical tests. The existing literature was heavily referenced, connecting their thorough study to multiple fields. The authors are to be commended for producing an exceedingly well-prepared manuscript – Figures and captions are clear, the document is brief and progresses naturally. There is little if anything I would take away from the text, and it has an appropriate level of complexity for the questions asked and answered. The document appears virtually free of typographical errors, and care was taken to produce pleasing figures that effectively illustrate the multiple notable findings.

Thank you.

After reviewing the manuscript I have a few minor issues that could be addressed to (in my personal view) enhance the study and are not intended to be critical of the manuscript as a whole. I have a few significant concerns that I believe should be addressed, but these could simply be due to a misreading of the manuscript and resolved by clarifying the appropriate result and methods sections. Once these issues are resolved I would be happy to recommend this article for publication.

Minor

Abstract, sentence 5: punctuation, "...visual cortex (V1), These populations show..."

Fixed.

Contour segmentation - Is there any relationship between distractor arrangement and target judgement? In the methods it is described that distractors are randomly set to 35deg or 55deg orientation, but one could imagine that given the fixed arrangement of the distractors, there are some combinations of distractors and targets which might induce higher-level perceptual effects – some arrangements might form curvilinear paths that are perceptually grouped into long contours, others might not. I'm just wondering if some of the difference between control and crowding might be explained by specific combinations of distractors. The authors note in the discussion that the spatial arrangement of distractors is significant to crowding, and I believe some arrangements of the distractors might in fact enhance discriminability. It seems that a simple analysis of psychophysical data collected in their experiments might capture this effect if it should exist.

We thank the reviewer for raising this point. Behaviorally, the monkeys (K, A, I) and matched human subjects (CAH, AG) saw 6 distinct distractor configurations with different arrangements of distractor orientations at positions relative to the target. All thresholds in this paper were fit separately within each fixed distractor arrangement, and reported values were averages from these individual fits (as stated in Methods).

To address the reviewer's suggestion that there may be different crowding across conditions, we compared thresholds for each arrangement. These are shown in Reviewer Figure 1A. One set (D1-D4, to

the left of the blue line) involved displays with a curvilinear or contour-like arrangement (indicated by the red line) of distractors and targets; the other set (D5,D6; to the right of the blue line) did not.

Subjects' threshold elevation under crowding varied slightly but not consistently with distractor arrangements. Importantly, all configurations elevated threshold; none reduced it. Moreover, there was no obvious or consistent change in crowding effect across conditions, or between displays with or without contour-like features. We conclude that there is no striking effect of distractor arrangement on the perceptual effects we report.

In our physiological measurements (and in perceptual measurement in the matched 2 human subjects CAH and AK), we only sampled 2 configurations of 8 distractors, which were both more akin to the discontinuous conditions (those with same sign of tilt at the end of the target). However, we did confirm that the change in target orientation information provided neuronal populations for these two conditions did not differ (paired t-test, $p=0.22$).

We have added a brief sentence to our discussion stating that the different distractor configurations induced similar perceptual and neuronal effects and that a proper assay of the influence of distractor configuration would require appropriately designed displays.

Page 10, paragraph 3 sentence 2: 'They' referring to 'our results' was a little awkward to parse and could be improved.

Fixed.

Additive vs multiplicative modulation: The additive facilitation and divisive scaling effect is clear and quite interesting. I believe that this result may be explained by a model where the neuron's activation function is nonlinear. Specifically, suppose response $R = f(r(\theta) + c)$, where $f(x)$ approximates an exponential from $-\infty < x < f^$, and is linear for $f^* < x$ and divisive scaling for $c < 0$ for some tuning curve shape r . However, this is likely just semantics describing the method by which a neuron's orientation preference is mapped into an output*

spike rate. It might be useful to mention why it matters that facilitation is additive and suppression multiplicative, and what mechanisms could be at play. It seems like the authors want to hint at suppression being the result of surround-dependent divisive inhibition, but this is a tenuous claim and should be avoided.

The comparison of additive vs multiplicative modulation was motivated by our attempt to understand how individual neurons could provide less orientation information when firing rates were elevated by distractors and also when they were suppressed. As argued in Fig 7, this requires those opposing modulations would have to have distinct functional forms. If facilitation had been multiplicative (and suppression had been subtractive) we would not have observed less information in the presence of distractors. We have modified our text to make this clearer.

There are many possible mechanisms that might contribute to the modulation we observe, including the form of response nonlinearity suggested by the reviewer. In our view, additive facilitation likely arises within the receptive field and would be expected from simple linear summation in a filter-based model of neuronal selectivity. The divisive suppression we report is also well-described in the literature (particularly for the surround, e.g. Cavanaugh et al., 2002; Coen-Cagli et al., 2015; both cited) though some studies have suggested that surround suppression is subtractive (Sceniak et al., 1999).

As the reviewer implies, our data were not designed to test mechanisms of modulation or to distinguish between subtractive vs multiplicative models of surround suppression. Testing these explanations would require sampling responses across a wide set of stimulus conditions than we explored. Our effort, instead, was to understand the contribution of V1 to perceptual crowding.

Major

Human and monkey psychophysics: eye position and feedback

I have a concern regarding the methods of measuring crowding between species. In your psychophysical methods you describe that humans had target presentations of 0.4 seconds, while monkeys had 0.25 seconds.

In the results paragraph 3 and Fig1c it is described that another experiment was run for humans matched to monkey psychophysics – does this include presentation times?

Yes, the additional human experiments used the same presentation duration as the monkey experiments. Thus, there are two sets of human measurements: one with 8 distractors and 400 ms presentations (matched to the neuronal recordings reported); and a second with 4 distractors and 250 ms presentations (matched to the monkey psychophysics). Both sets of experiments revealed similar crowding. This has been clarified in Results and Methods.

It might be helpful to explicitly title the left and right regions of Fig1c to separate these two conditions.

Done.

In any event, 0.25 seconds is quite long for a human judgement, long enough for a visually-guided saccade and spatial attention to shift. In the monkey case eye position was monitored, but it's unclear that human eye positions were similarly restricted. Given that eye-tracking details were given in the primate psychophysics paragraph of the methods, one can assume that eye position was not monitored for human subjects. I believe that it is necessary to ensure subjects were fixating within a 1 deg. window to eliminate the possibility of voluntary or involuntary saccades contaminating performance.

Humans were instructed to maintain strict fixation on a small fixation point, but eye position was not tracked. To address concerns about eye movements influencing our measurements of human crowding, we re-ran two subjects (8 distractors, 400 ms presentation), using the same eye tracking approach as used for the monkeys. Reviewer Figure 2 shows discrimination thresholds and threshold elevations

under crowding for both subjects for all trials ('all') as well as for the subset of trials in which the eyes remained within a 1 deg diameter fixation window ('fix'). Both thresholds and threshold elevation under crowding were similar in trials with successful fixation. Further, threshold elevations were significantly greater than 1 in both cases, and similar to those previously reported in our manuscript (with no eye tracking). We note that eye tracking in human studies of crowding is rarely done, so this represents a stronger experimental control than the standard in the literature.

We have updated the methods section to report that our measurements in human subjects are unlikely to be affected by eye movements.

Also, it is not mentioned if humans received feedback for correct/incorrect choices – the monkeys were provided feedback by water reward, and therefore could adapt their decision strategy to maximize performance. I believe that human subjects would also need to be given feedback after every trial to ensure a fair comparison. If either or both of these conditions were matched across humans and animals, please indicate so within the text. A difference in these conditions would need to be justified.

Humans also received feedback for correct/incorrect choices, in the form of an auditory tone on incorrect trials. The methods have been updated to state this.

Phase

I am concerned about the lack of phase randomization within the psychophysical experiments. Given that the targets were presented at a fixed spatial frequency and phase, I believe it is possible for an observer to perform the task by instantaneously measuring a single pixel in the display – it being above or below 50% gray would indicate horizontal or vertical orientation. I understand that this would not be possible for an animal when viewing a target at 4-5 degrees eccentricity, but I still wonder why phase wasn't randomized during monkey

psychophysics as simple cues to the judgement exist in the stimuli. I feel it is necessary for the authors to justify this difference in experimental conditions.

We thank the reviewer for raising this concern and lack of clear motivation. We did not phase randomize stimuli in monkey perceptual experiments so that we could later perform neuron-choice analyses of recorded neuronal data on the crowding task (i.e. which require many repeats of identical stimuli). We note that in order to implement the reviewer's suggested behavioral strategy, monkeys would have to monitor a pixel value at a fixed time point, since the gratings we presented were drifting. Further, we did confirm in control sessions in one animal that randomizing phase had no influence on task performance or the strength of perceptual crowding.

Given this, and the similar magnitude of human (phase randomized) and monkey (not phase randomized) crowding, it seems unlikely that phase randomization (or its absence) strongly affects crowding magnitude. We have updated the text to justify not varying starting spatial phase in the monkey perceptual experiments.

Response dynamics

I would expect that there is an appreciable latency in the response modulation due to crowding, should the multiplicative scaling of responses be due to connections within V1. Therefore, how does the modulation of responses demonstrated in Fig7 evolve over time? If the neural response to crowding is delayed, I would expect that the V1 populations would more accurately decode orientation during the initial transient response period, and get worse as crowding effects modulate responses. It might be the case that V1 could accurately decode orientation during this initial transient period and thus not limit feature representation under crowding. I feel that it is critical for the authors to address this to rule out crowding as merely a latent effect within V1, as that would place the neural basis for crowding at a later stage of processing

To address this helpful suggestion, we redid our main decoding analysis at finer temporal resolution. We measured population activity in 100ms windows, a window chosen to be small enough for better temporal resolution but still large enough to provide sufficient spiking activity for reasonable decoding performance. This was done at 3 different times relative to stimulus onset: an initial window of 50-149 ms (to account for approximate average neuronal response onset latency), and additional windows of 150-249 ms and 250-349 ms. In each epoch, we computed the response modulation (response to target with distractors compared to targets alone). This revealed robust suppression in all 3 epochs though the suppression was weaker in the first epoch (Reviewer Figure 3A, as predicted by the reviewer). For each time window, we then calculated population neurometric functions for targets alone and targets with distractors, as in the main text. Reviewer Figure 3B shows the change in population threshold: distractors caused an elevation in threshold in each epoch.

Because these analyses did not reveal compelling dynamics of the results we report in the main text, we have not added them to our revised manuscript. We are also reluctant to attempt any inference of the role of feedback based on measurements of dynamics. Even if the loss of target orientation information had only been apparent after a long delay—not the case in Reviewer Figure 3—one would still not know whether was because the effects arose from feedback from higher cortex or dynamics within V1 (e.g. as in the work of Ken Miller; Rubin et al., 2015, Neuron 85: 402, Hennequin et al., 2018, Neuron 98: 846). Of course, if the reviewer or editor feels strongly that we should touch on this issue in the manuscript, we would be happy to do so.

Reviewer #2 (Remarks to the Author):

Overall this is a well-written manuscript investigating how contextual effects in primary visual cortex might underlie the psychophysical phenomenon of crowding. As the authors clearly articulate, crowding has been studied extensively in human subjects but the neural basis has not been systematically investigated. This manuscript shows that monkeys, like humans, experience crowding, and details how contextual effects in V1 could at least partially underlie crowding. This manuscript and data represent a valuable, novel addition to the literature. Experiments are well-designed and the analyses are appropriate. All of my comments below are suggestions aimed at trying to improve clarity and detail.

Thank you for the supportive remarks.

1. It will be beneficial to the reader if the authors could provide more details in methods specifically with regard to the rationale behind parameter choice. I do appreciate these are primate experiments and the parameter space cannot be explored extensively but some more detail would be appreciated. For example, the stimulus eccentricity of 4.24 for the behavioral task – was this based on the mean RF in the monkey experiments?

We apologize for the lack of clear motivation in experimental choices. The stimulus eccentricity was chosen to be the approximate RF position of planned future neuronal recordings.

Is there are reason for choosing 400 ms and 8 distractors for the human studies but 250 ms and 4 distractors for the monkey? Why is target-distractor spacing 1.04 in one experiment but 1.1 in the other? It seems some of this is motivated by another study but a little more detail as to why those specific parameters were chosen

would be beneficial to the reader. As of now, these seem rather arbitrary. It seems that a short paragraph explaining why these numbers were chosen would be helpful.

The human perceptual experiments used two different sets of parameters. One set of humans was run with 8 distractors and 400 ms presentation duration, to match the anesthetized physiology recordings that are the focus of this manuscript. A second set of humans was run with 4 distractors and 250 ms presentation duration, to match the monkey perceptual data. For these monkey perceptual experiments, we used 4 distractors (rather than 8) so that if needed we could place distractors directly adjacent to the target, without distractors overlapping each other (i.e. if we wanted to strengthen crowding further). We used a briefer presentation time in the monkey perceptual work (250 ms instead of 400 ms) to facilitate neuron-choice analysis (to be reported elsewhere). Previous work has shown that longer presentation times complicate efforts to relate neuronal responses to choices on a trial-by-trial basis, because it is unclear which response epoch is used to guide behavior. By using shorter presentation durations, we reduce this ambiguity.

We have changed Figure 1 to help clarify the different paradigms, and added additional explanatory text to Methods to motivate our choices.

2. Were eye signals monitored in the experiment with the human subjects? Please provide details including window size.

Eye signals were not monitored in human subjects, for the measurements reported in the original manuscript. To address any influence of eye movements, we ran additional sessions in 2 subjects with eye tracking, yielding similar results. Please see response to Reviewer #1 (Reviewer Figure 2) for further details.

3. Is it true that the distractors orientations all deviated by the same amount from 45 in the monkey task as in the human task? If so, please state explicitly.

The deviation of the distractors from the target was fixed within a day's session, but was occasionally varied across sessions (range 2-30 degrees; median, 5 degrees). Humans were run with 10 degree deviation, which is in the range of the monkey deviations and is matched to the physiological data. The methods section has been updated to reflect this.

4. In Figure 1a it would be helpful to show panels for both types of tasks (human and monkey) and then in 1c, show small versions of those corresponding to the two human and one monkey panel. As presented, the figure and legend don't provide any hints about the differences between the two human panels. Help the reader!'

We apologize for the confusion. We have updated the figure with additional text to better clarify the differences among the perceptual experiments.

5. In Fig 2a it will be good to visualize the size of V1 RFs. So, the stimuli (shown in red) could simply be drawn as outlines so the V1 RFs show through.

We have changed the stimuli to outlines to better highlight the RFs as the reviewer suggested.

6. How are V1 RF sizes estimated? Is it 1SD? 2SDs? Please specify.

We have updated the figure legend to explain that the RF extent represents 2 SDs of a Gaussian fit to the neuronal responses.

7. Are units in Fig 2c included in 2b? If so, please identify.

Units in 2c are different from those in 2b. This is now reported in the figure legend.

8. *Levi and Carney 2009 is cited as Carney and Levi on page 10. Either that, or a reference is missing. Please fix.*

We thank the reviewer for catching this. The reference is Levi and Carney. Citations in the text are now numeric, as per Nature Communications requested style.

Reviewer #3:

In this study, Henry and Kohn investigate the neurophysiological underpinnings of visual crowding. Crowding is ubiquitous in peripheral vision and although much is known about the phenomenology, almost nothing is known about the underlying neural mechanisms. This study is important and is the first to directly assess the possible neural mechanisms and associated computations in the primary visual cortex (V1). The authors conclude that spatial contextual effects limit the discriminability of visual features in V1 population activity and this loss of information might provide a fundamental limiting factor for the rest of the visual processing hierarchy.

I would like to congratulate the authors for an elegantly conducted study and for a very well written manuscript.

Thank you for these kind comments.

I have no major comments.

I have two minor comments:

— the criteria that were used to estimate the discrimination thresholds from the psychometric curves is not stated

Thank for catching this omission. We define the threshold as 75% correct and now state this in the Methods.

Note that in Figure 1 of the original submission, the thresholds for the psychometric curves were accidentally defined as 1 s.d. of the cumulative normal fits to the data (e.g. 84% correct). Figure 1 and the corresponding values in the main text have now been updated to reflect a 75% criterion for threshold (as used in the physiology). Though the values changed slightly, the results and conclusions are the same.

*— the Nandy & Tjan (2012) study has been mischaracterized (p3 and p10) in stating that crowding can be attributed to saccadic sampling. That study explored saccadic sampling as a means to explain the radially elongated **shape** of crowding zones but was very much in alignment with the present study that crowding itself was caused by contextual effects.*

We apologize for misstating the conclusions of Nandy & Tjan (2012); this was entirely unintentional. We have updated the text to correct this error.

Anirvan Nandy
Dept. of Neuroscience
Yale University

Reviewers' Comments:

Reviewer #1:

Remarks to the Author:

The authors have thoroughly addressed my comments and concerns. I have no outstanding issues with this manuscript and strongly recommend it for publication.

Reviewer #2:

Remarks to the Author:

All my comments are addressed. I have no further comments.

Reviewer #3:

Remarks to the Author:

I have no further comments. I am happy to recommend this article for publication.

[For the future, I would urge the authors to submit revised documents with changes highlighted which will make the review process much quicker]

Anirvan Nandy, PhD.
Yale University

REVIEWERS' COMMENTS:

Reviewer #1 (Remarks to the Author):

The authors have thoroughly addressed my comments and concerns. I have no outstanding issues with this manuscript and strongly recommend it for publication.

Reviewer #2 (Remarks to the Author):

All my comments are addressed. I have no further comments.

Reviewer #3 (Remarks to the Author):

I have no further comments. I am happy to recommend this article for publication.

[For the future, I would urge the authors to submit revised documents with changes highlighted which will make the review process much quicker]

Anirvan Nandy, PhD.
Yale University